# Evaluation of Zonulin Expression and Its Potential Clinical Significance in Glioblastoma

**DOI:** 10.3390/cancers16020356

**Published:** 2024-01-14

**Authors:** Roberta Repossi, Rita Martín-Ramírez, Fuensanta Gómez-Bernal, Lilian Medina, Helga Fariña-Jerónimo, Rebeca González-Fernández, Pablo Martín-Vasallo, Julio Plata-Bello

**Affiliations:** 1Neurogenetics of Rare Disease Group, Hospital Sant Joan de Déu, 08950 Barcelona, Spain; 2Clinical Neuroscience Research Group, University of La Laguna, 38320 La Laguna, Spain; 3Department of Molecular Biology, Faculty of Biology, University of La Laguna, 38320 La Laguna, Spain; 4Department of Biochemistry, Hospital Universitario de Canarias, 38320 S/C de Tenerife, Spain; 5Department of Neurosurgery, Hospital Universitario de Canarias, 38320 S/C de Tenerife, Spain

**Keywords:** glioblastoma, zonulin, prognosis, blood–brain barrier, glioma stem cell (GSC)

## Abstract

**Simple Summary:**

Glioblastoma, the most deadly type of brain tumor in adults, poses a significant treatment challenge and often leads to poor outcomes. A recent study delved into the role of a protein called zonulin, known for its impact on the body’s barrier functions, in glioblastoma. The research involved examining zonulin levels in both the bloodstream and tumor tissues of 34 newly diagnosed glioblastoma patients. The study discovered that higher levels of zonulin in tumors were linked to increased aggressiveness, which in turn correlated with a poorer prognosis. Elevated serum levels of zonulin in the blood also indicated a connection with a less favorable prognosis. In laboratory experiments, glioblastoma cells expressed higher levels of zonulin, particularly under conditions resembling glioma stem cells, which are associated with treatment resistance. This research suggests that heightened zonulin levels in both tumors and the blood may signal a more challenging prognosis for glioblastoma patients, underscoring the potential importance of this protein in the disease.

**Abstract:**

Glioblastoma, the deadliest adult brain tumor, poses a significant therapeutic challenge with a dismal prognosis despite current treatments. Zonulin, a protein influencing tight junctions and barrier functions, has gained attention for its diverse roles in various diseases. This study aimed to preliminarily analyze the circulating and tumor zonulin levels, evaluating their impact on disease prognosis and clinical–radiological factors. Additionally, we investigated in vitro zonulin expression in different glioblastoma cell lines under two different conditions. The study comprised 34 newly diagnosed glioblastoma patients, with blood samples collected before treatment for zonulin and haptoglobin analysis. Tumor tissue samples from 21 patients were obtained for zonulin expression. Clinical, molecular, and radiological data were collected, and zonulin protein levels were assessed using ELISA and Western blot techniques. Furthermore, zonulin expression was analyzed in vitro in three glioblastoma cell lines cultured under standard and glioma-stem-cell (GSC)-specific conditions. High zonulin expression in glioblastoma tumors correlated with larger preoperative contrast enhancement and edema volumes. Patients with high zonulin levels showed a poorer prognosis (progression-free survival [PFS]). Similarly, elevated serum levels of zonulin were associated with a trend of shorter PFS. Higher haptoglobin levels correlated with MGMT methylation and longer PFS. In vitro, glioblastoma cell lines expressed zonulin under standard cell culture conditions, with increased expression in tumorsphere-specific conditions. Elevated zonulin levels in both the tumor and serum of glioblastoma patients were linked to a poorer prognosis and radiological signs of increased disruption of the blood–brain barrier. In vitro, zonulin expression exhibited a significant increase in tumorspheres.

## 1. Introduction

Glioblastoma is the most common and aggressive primary brain tumor in adults, with an average age at diagnosis of 65. It is classified as Grade IV by the World Health Organization (WHO) due to its high malignancy and invasive infiltration into surrounding brain tissue [1]. With an annual incidence of approximately 5 cases per 100,000 inhabitants, it more frequently affects males than females [2]. Despite standard treatments, including surgery, radiotherapy, and chemotherapy (temozolomide), patients diagnosed with glioblastoma often face a bleak prognosis, with a median survival of around 15 months [3]. Consequently, there is a critical need to identify new therapeutic strategies and targets to improve the prognosis of this disease.

Zonulin, originally identified as pre-haptoglobulin-2, is a secreted protein that modulates the permeability of tight junctions between epithelial and endothelial cells, influencing important barrier functions, such as the intestinal barrier [4,5]. Zonulin is responsible for the regulation of tight junctions (TJs) and the homeostasis of barriers [6] and can induce the reversible disassembly of TJs by binding to the epidermal growth factor receptor (EGFR) via PAR-2, leading to calcium-dependent actin polymerization [7]. The study of zonulin has gained attention in recent years due to its involvement in various diseases, especially in the context of gastrointestinal and autoimmune disorders [8,9]. The dysregulation of zonulin is associated with immune-mediated diseases, such as inflammatory bowel disease [10], celiac disease [4], and rheumatoid arthritis [9], among others, suggesting a critical role in gastrointestinal barrier homeostasis. However, this protein also seems to have implications in cerebral physiology. Emerging studies suggest that zonulin may play a crucial role in the pathogenesis of various neurological diseases. Altered levels of zonulin are associated with neuroinflammatory disorders, such as multiple sclerosis [11], and it has been postulated that the dysfunction of this protein may contribute to the abnormal permeability of the blood–brain barrier (BBB) [12], a central event in the progression of various brain pathologies. Additionally, preliminary research suggests that zonulin could influence neurodegeneration as a factor that could have some impact on diseases such as Alzheimer’s [13] and Parkinson’s [14]. Zonulin levels have also been evaluated in patients with glioblastoma. In this regard, Naryzhny S et al. (2021) identified elevated serum levels of zonulin in patients with glioblastoma [15]. Furthermore, glial tumor cells seem to express this protein [15,16], and its expression level is associated with the malignancy grade that the glial tumor shows, as well as with the disruption of the BBB [16]. In this sense, a correlation between zonulin levels and the degradation of the blood–brain barrier was described in an in vitro migration study, favoring the increased mobility of glioma stem cells [17].

However, despite these initial findings, a more exhaustive analysis is needed for the understanding of the complete impact of zonulin on glioblastoma biology and prognosis. In this regard, it would be interesting to simultaneously analyze zonulin levels in plasma and tumor samples from patients with glioblastoma, as well as their relationship with the clinical and radiological characteristics of these patients and their prognosis. On the other hand, it is essential to confirm that glioblastoma tumor cells produce this protein and to better characterize the aspects that may modulate such production.

Therefore, the objective of this study is to conduct a preliminary analysis of the relationship that circulating and tumor zonulin levels may have with the prognosis of the disease in terms of progression-free survival (PFS), as well as with other clinical–radiological factors. Additionally, the study aims to analyze the in vitro expression of zonulin in different cell lines cultivated under different conditions.

## 2. Methods

### 2.1. Patients and Data Collection

A total of 34 newly diagnosed glioblastoma patients (Isocitrate Dehydrogenase [IDH] 1&2 wildtype) were included from January 2021 to December 2022. In 27 out of the 34 patients (Appendix A), a blood sample was collected at 8:00 am before the initiation of corticosteroid treatment (commonly used in this patient population) and prior to surgery. Serum was centrifugated and stored at −80 °C until measurements. Zonulin serum levels were measured with the ELISA method using a Human Zonulin ELISA Kit (Catalog No. EEL-H5560; Elabscience Biotechnology Inc., Houston, TX, USA) in line with the manufacturer’s instructions. Haptoglobin serum levels were measured by immunoturbidimetry at Cobas 801Roche^®^.

For 21 out of the 34 patients (Appendix A), an excess tumor tissue sample beyond what was required for pathological diagnosis was obtained and preserved in Trizol^®^ (Catalog No. 155960; Invitrogen, Waltham, MA, USA) at −80 °C for further analysis of zonulin expression. Both blood and tumor samples were available for 14 out of the 34 patients.

Clinical data were retrieved from the medical records of each patient. Molecular data from the tumor sample (methylation of the methyl-guanil-methyl-transferase promoter and Ki-67) were extracted from the pathological anatomy report. Preoperative magnetic resonance imaging (MRI) was qualitatively and quantitatively analyzed. The Oncohabitats platform (https://www.oncohabitats.upv.es/ accessed on May–June 2023) was employed to segment the tumor lesion and obtain quantitative data on necrosis, contrast enhancement, and edema volumes.

### 2.2. Ethics

The study received approval from the clinical research ethics committee at our institution. The study was conducted in accordance with the principles outlined in the Declaration of Helsinki. Prior to inclusion in the study, all patients provided informed consent by signing a consent form.

### 2.3. Cell Culture and Tumorsphere Induction

U87MG, U118MG, and A172 human high-grade glioma cell lines were used purchased from ATCC (Catalog No. HTB-14, HTB-15 and CRL-1620, respectively; Manassas, VA, USA). Cell lines were seeded in T75 flasks at a seeding density of 1,000,000 cells, in standard medium (SM). SM was prepared as follows: Dulbecco’s modified Eagle’s medium/Ham’s F-12 (DMEM/F12, Catalog No. 11330032; Gibco, Thermo Fisher Scientific, Waltham, MA, USA), containing L-Glutamine and 15 mM HEPES, supplemented with 10% fetal bovine serum (FBS, Catalog No. S1810; Biowest, Nuaillé, France), 59 mg/L Penicillin, 100 mg/L Streptomycin, and 25 mg/L Amphotericin (Catalog No. P3032, S9137 and A2411; Sigma-aldrich, San Luis, MO, USA). Cells grew in monolayers with a humidified atmosphere, with 5% CO_2_ at 37 °C. Medium was refreshed every two days. Upon reaching confluence, cells were treated with trypsin-EDTA (Catalog No. L0930; Sigma-aldrich, San Luis, MO, USA) and seeded at a concentration of approximately 100,000 cells/mL in 35 mm low attachment plates (Nuclon Sphera, Catalog No. 174943; Thermo Fisher Scientific, Waltham, MA, USA) with SM or tumorsphere medium. GSC medium was prepared according to [18,19] as follows: DMEM/F12, containing L-Glutamine and 15 mM HEPES (Catalog No. 11330032; Gibco, Thermo Fisher Scientific, Waltham, MA, USA), supplemented with antibiotics (59 mg/L Penicillin, 100 mg/L Streptomycin, and 25 mg/L Amphotericin; Catalog No. P3032, S9137, and A2411; Sigma-aldrich, San Luis, MO, USA), recombinant human epidermal growth factor (rhEGF, Catalog No. 236-EG-200; R&D systems, McKinley Place NE, MN, USA) 20 ng/mL, basic fibroblast growth factor (bFGF, Catalog No. 233-FB-025/CF; R&D systems, McKinley Place NE, MN, USA) 20 ng/mL, and B-27 additive 1X (Catalog No. A3353501; Gibco, Thermo Fisher Scientific, Waltham, MA, USA). Gliomasphere formation was monitored daily, and pictures of the same field were taken at 10× magnification using an OLYMPUS CKX53 microscope with a DP23 digital camera and the associated imaging software, cellSens 4.1. Pellets were collected after 72 h of culture. All experiments were performed in triplicate.

### 2.4. Western Blot

Zonulin protein levels were analyzed by Western blotting. Briefly, tissue samples were minced in small pieces of 2 mm and frozen in liquid nitrogen. Then, fragments were disaggregated with a mortar and a pestle. Homogenized tissues and cell pellets were directly resuspended in Laemmli sample buffer (LSB) 1x. The same protein volumes were separated in polyacrylamide gels and transferred onto polyvinylidene fluoride membranes (Catalog No. PVH00010; Millipore, Burlington, MA, USA). Primary antibody against zonulin (dilution 1:500, Abbexa, Cat #abx109849) was used. HRP-conjugated anti-rabbit (dilution, 1:50,000, Catalog No. F9887; Sigma-aldrich, San Luis, MO, USA) was used as secondary antibody. Membranes were incubated with chemiluminescent reagent following the manufacturer’s instructions (Catalog No, WBKLS0100; Millipore, Burlington, MA, USA). Images were visualized using ChemiDoc equipment (BIORAD) and analyzed by Image Lab 6.1 software (BIORAD). Finally, protein levels were quantified by total protein normalization after membrane staining with Coomassie Brilliant Blue.

### 2.5. Statistical Analysis

Three comparative analyses were conducted on the distribution of clinical–radiological and molecular variables of interest: (1) between patients with low and high zonulin expression in the tumor sample; (2) between patients with low and high serum levels of zonulin; and (3) between patients with low and high serum levels of haptoglobin. In both cases, the cut-off point used to categorize patients was the median expression of zonulin in tumor samples (p50 = 406.79) and the median levels of serum zonulin (p50 = 315.0 ng/mL) as well as haptoglobin (p50 = 212.0 mg/dL).

Non-parametric statistical tests were employed for the comparative analysis. Additionally, a comparison of zonulin and haptoglobin expression groups in relation to progression-free survival (PFS) was conducted using the Log-Rank test and Kaplan–Meier curves.

Moreover, differential zonulin expression in glioblastoma cell lines cultured in standard medium compared to one specific for tumorspheres was analyzed. A non-parametric statistical test (Mann–Whitney U test) was employed, considering statistical significance when *p* < 0.05.

## 3. Results

### 3.1. Analysis of Preoperative Serum Levels of Zonulin and Haptoglobin in Glioblastoma Patients

The determination of zonulin and haptoglobin levels was performed on serum samples from glioblastoma patients before surgery and without prior exposure to steroid treatment. The mean levels of zonulin were 454.18 ng/mL (SD = 413.76), while those of haptoglobin were 208.83 mg/dL (SD = 85.42). The serum levels of both proteins showed no correlation (Spearman correlation coefficient = −0.010; *p* = 0.961). Similar to the analysis of zonulin levels in tumor samples, the median of serum zonulin levels (p50 = 315.0 ng/mL) was used to create two groups of circulating zonulin levels: high (>p50) and low (≤p50). Comparative analysis between both patient groups revealed a different gender distribution (more males in the low-zonulin-level group, *p* = 0.000) and a higher mean age in the high-zonulin-level group (68.1 vs. 60.36 years, *p* = 0.046) (Table 1). The remaining clinical and molecular variables analyzed did not show differences between the two groups. However, it is noteworthy that there was a positive trend favoring low zonulin expression regarding PFS. In this regard, patients with low levels of zonulin had a superior PFS compared to patients with high expression levels (10.5 vs. 3.7 months; *p* = 0.1) (Table 1; Figure 1).

Finally, using the median of circulating haptoglobin levels (p50 = 212.0 mg/dL), a comparative analysis was conducted between the group of patients with high (> p50) and low (≤p50) circulating haptoglobin levels. A higher proportion of MGMT methylated patients were categorized as having high haptoglobin levels (78.6% vs. 33.3%; *p* = 0.045). No other significant differences were observed in the distribution of the rest of the clinical–radiological and molecular variables, except for a trend in PFS, with better prognosis observed in patients with high circulating haptoglobin levels (15.0 vs. 4.5 months; *p* = 0.1) (Appendix A, Appendix A).

### 3.2. Clinical–Radiological Differences between Patients with High and Low Expression of Zonulin in the Tumor Sample

An analysis of zonulin expression was conducted on samples of tumor tissues, using the median (p50 = 406.79) to compare patients with high (>p50) and low (≤p50) zonulin expression in the tumor (Table 2). Patients with high zonulin expression exhibited a larger volume of both contrast enhancement (28.13 vs. 12.91 cc; *p* = 0.020) (Figure 2) and edema (79.78 vs. 42.44 cc; *p* = 0.037) (Figure 1) in preoperative MRI. Additionally, patients with high zonulin expression displayed a more heterogeneous contrast enhancement (80.0% vs. 45.5%), although this difference did not reach statistical significance (*p* = 0.183).

The remaining clinical and molecular variables analyzed did not show differences between the two groups, but survival analysis demonstrated a longer progression-free survival (PFS) in the low-zonulin-level group compared to the high-level group (6.6 vs. 3.3 months; *p* = 0.024) (Table 2, Figure 1).

### 3.3. In Vitro Zonulin Expression in Glioblastoma Cell Lines

Western blot was performed on three glioblastoma cell lines (U87MG, U118MG, and A172) to detect zonulin under two different experimental conditions: one considered standard and another specific for the generation of tumorspheres. All three cell lines exhibited zonulin expression under standard conditions, with no significant differences among them (Kruskal–Wallis, *p* = 0.694). This expression increased significantly when cells were cultured in a medium and on a plate specifically designed for tumorspheres (Mann–Whitney U; U87MG, *p* = 0.040; U118MG, *p* = 0.025; and A172, *p* = 0.083) (Figure 3, Appendix A). The expression of zonulin in GSC cultures did not show significant differences based on cell type (Kruskal–Wallis, *p* = 0.119).

## 4. Discussion

This study aimed to explore the relationship between the clinical–radiological characteristics of glioblastoma patients and zonulin expression, both within the tumor and in serum levels. Zonulin is a haptoglobin precursor protein responsible for increasing permeability in certain barriers in our body, with proven expression in glial tumors. In this work, tumor zonulin expression correlated with a poorer prognosis and increased blood–brain barrier (BBB) disruption. Elevated circulating zonulin levels were associated with an unfavorable prognosis, demonstrating reduced progression-free survival (PFS) in this patient group. To confirm glioblastoma tumor cells’ capability to produce this protein, in vitro assays were conducted on cell lines, revealing heightened zonulin expression in tumorspheres. We discuss the main findings and implications of the study.

Both patients with increased zonulin expression in the tumor sample (Figure 2A) and those with higher serum levels (Figure 2B) demonstrated a clear trend with a worse PFS. This finding is novel, representing the first reported association between glioblastoma prognosis and zonulin expression levels. These results align with previous studies describing increased zonulin expression in glial tumors as the malignancy grade rises [16].

In any case, the lack of correlation between tumors and circulating zonulin expression suggests that, aside from tumor production, circulating zonulin may have other origins, with the intestinal source being the most plausible and supported by previous evidence [4,20]. Steroids, commonly used in glioblastoma patients for cerebral edema control, seem to play a role in modulating intestinal microbiota [21], contributing to zonulin production. Some authors have proposed a relationship between dysregulation of the microbiota and intestinal barrier as facilitators of glioblastoma development [22,23]. Therefore, conducting further studies investigating the intestinal microbiota in glioblastoma patients would be intriguing to analyze its influence on zonulin production.

However, despite the lack of correlation between tumors and circulating zonulin, it does not exclude the possibility that zonulin detected in a tumor originates from the periphery, i.e., serum levels. To confirm glioblastoma tumor cells’ production of this protein, an assay was conducted on three glioblastoma cell lines, demonstrating expression when cultured under standard conditions that allow for the development of differentiated astrocytic tumoral cells. It is interesting to note a significant increase in zonulin expression in all lines when cells were cultured in serum-free tumorsphere medium. Tumorspheres are supposed to be formed by glioma stem cells (GSCs), which are significant within glioblastoma and play a crucial role in development and growth, conferring resistance to standard treatments [24,25]. In addition to their potential to support tumor development, GSCs are associated with BBB disruption, are typically located in perivascular niches, create crucial interactions with endothelial cells [26,27], and promote neoangiogenesis [28,29,30]. Zonulin expression by GSCs would facilitate their migration capability [17] and contribute to BBB disruption [16]. Our study findings, demonstrating increased edema volume and contrast uptake (both indicators of BBB disruption) in the high-zonulin-expression patient group, support this explanation, emphasizing the potential role of zonulin in two fundamental aspects of this disease: BBB disruption with edema development and the consequent neurological symptoms, and resistance to treatments associated with earlier disease progression. Nevertheless, further studies are needed to delve into this potential role of zonulin in glioblastoma biology.

Finally, it is noteworthy that circulating haptoglobin levels exhibited an opposing effect to zonulin; high levels were associated with a clear trend of better PFS. While haptoglobin has been proposed as a potential blood biomarker for monitoring glioblastoma patients, previous works analyzing this aspect indicated zonulin as the proteoform increased in the blood of glioblastoma patients [15,31]. It appears that in cases where zonulin is processed and forms the mature protein, it loses its pathogenic capacity. Understanding the mechanisms involved in haptoglobin assembly could help identify molecular targets for reducing zonulin levels and improving glioblastoma patient prognosis.

### Limitations

The main limitation of this study is the sample size, underscoring the need for future research with more extensive cohorts to validate the relationship between zonulin expression, prognosis, and BBB disruption in glioblastoma. Additionally, it would be beneficial to confirm, in future research, the findings related to the increased expression of zonulin under pluripotent conditions in primary GSC lines and patient-derived tumorspheres.

## 5. Conclusions

Increased zonulin expression in glioblastomas, both locally and systemically, correlates with an unfavorable prognosis. Conversely, elevated haptoglobin levels, derived from zonulin, are associated with a more favorable prognosis. In vitro experiments confirm significant zonulin expression in glioblastoma stem cells, indicating its potential impact on disease progression.

## Figures and Tables

**Figure 1 cancers-16-00356-f001:**
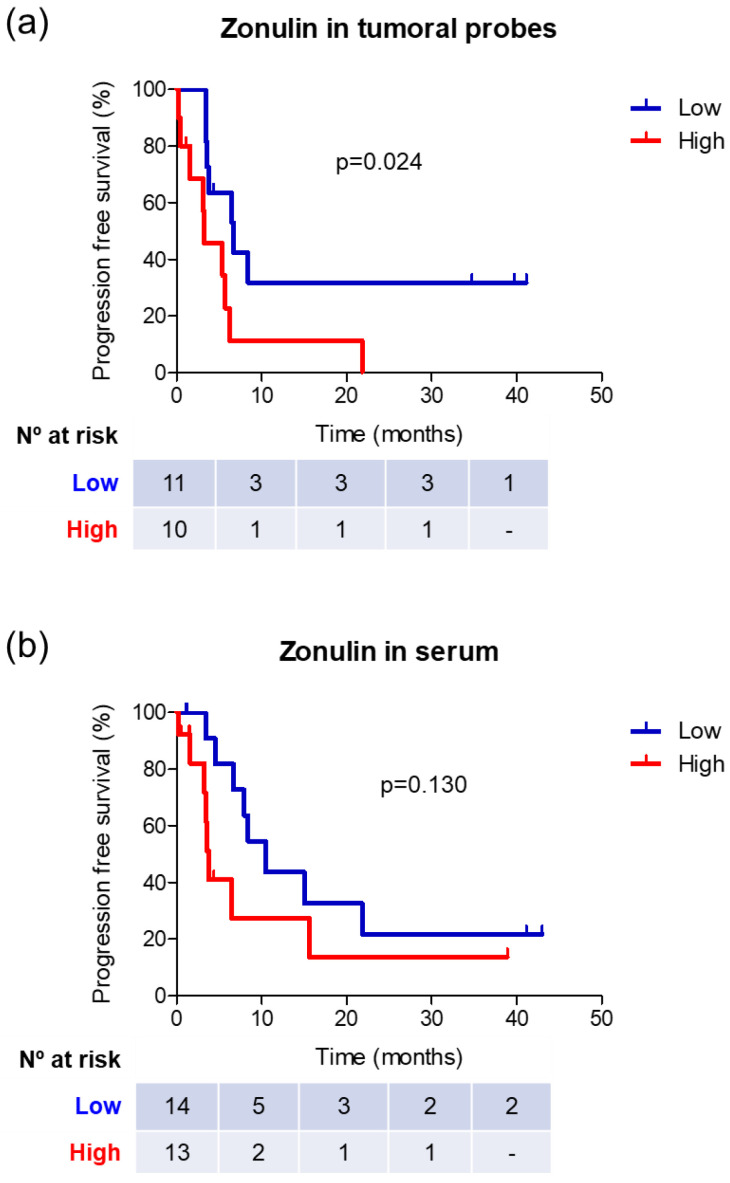
Survival analysis in different groups of zonulin expression at the tumor level (**a**) and serum level (**b**).

**Figure 2 cancers-16-00356-f002:**
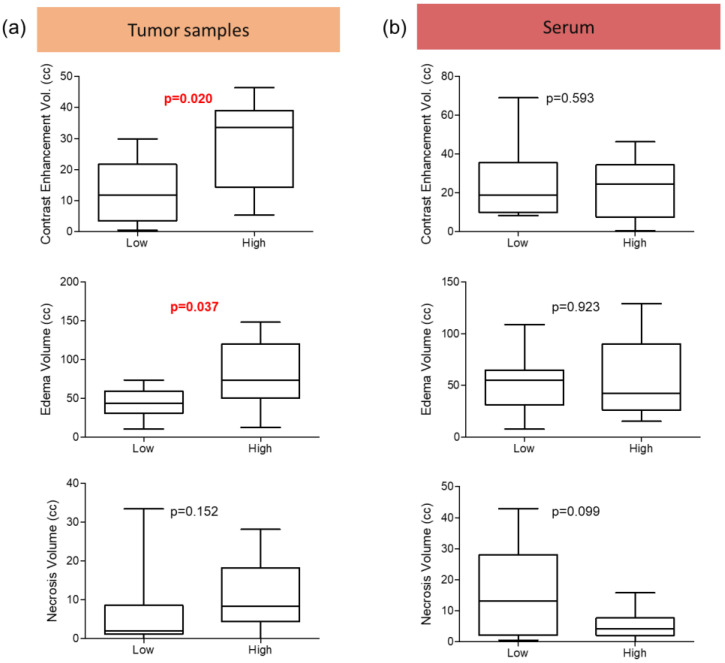
Distribution of preoperative magnetic resonance imaging tumor segmentation data in relation to the level of zonulin expression in the tumor (**a**) or serum levels (**b**). *p*-value in red indicates statistical significance.

**Figure 3 cancers-16-00356-f003:**
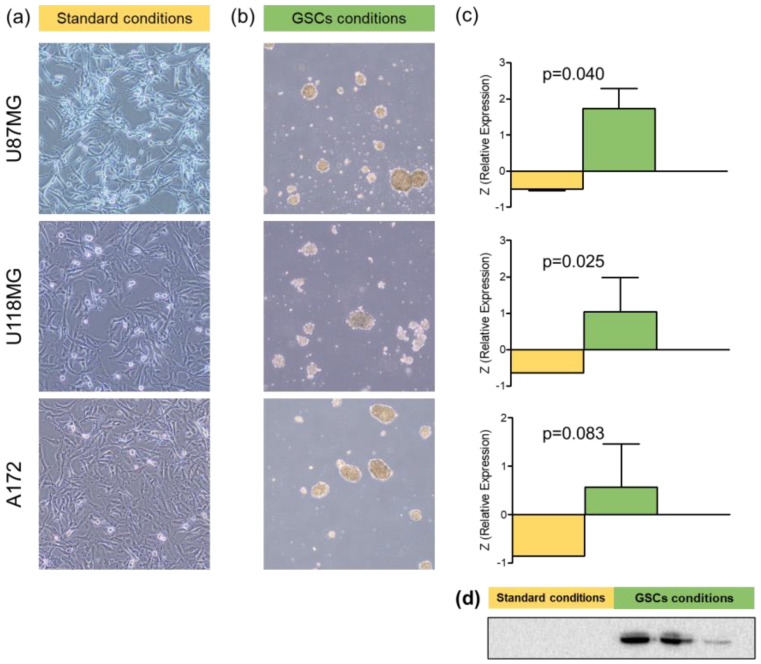
Zonulin expression in glioblastoma cell line cultures under different conditions**.** (**a**) Appearance of cultures of different cell lines in standard medium and adherent support at 96 h; (**b**) appearance of cultures of different cell lines in medium specific for tumorspheres and non-adherent support at 72 h; (**c**) zonulin expression in different cell lines under the two culture conditions. Bars represent the mean expression, and whiskers indicate the standard deviation; (**d**) visual depiction of Western Blot results in three glioblastoma cell lines under different culture conditions.

**Table 1 cancers-16-00356-t001:** Comparison between groups of high vs. low zonulin serum levels.

Variable		Low Zonulin (*n* = 14)	High Zonulin (*n* = 13)	*p*-Value
Age (years)		60.36 (SD = 12.68)	68.1 (SD = 6.37)	0.046 ^a^
Gender (male/female)		13:1	1:12	0.000 ^b^
Karnofsky < 70		1 (7.1%)	2 (15.4%)	0.596 ^b^
Contrast enhancement	Periferic	9 (64.3%)	6 (46.2%)	0.449 ^b^
	Heterogeneous	5 (35.7%)	7 (53.8%)	
Contrast enhancement volume (cc)		24.74 (SD = 18.06)	20.99 (SD = 14.62)	0.593 ^a^
Edema volume (cc)		51.15 (SD = 28.36)	55.37 (SD = 38.13)	0.923 ^a^
Necrosis volume (cc)		15.63 (SD = 13.94)	5.37 (SD = 4.31)	0.099 ^a^
Resection	Partial	2 (14.3%)	6 (46.2%)	0.132 ^c^
	Subtotal	4 (28.6%)	1 (7.7%)	
	Total	8 (57.1%)	6 (46.2%)	
Ki67 (%)		28.54 (SD = 13.91)	21.0 (SD = 12.79)	0.181 ^a^
MGMT methylation		7 (50.0%)	8 (66.7%)	0.453 ^b^
Progression-free survival (months)		10.5 [6.6–14.3]	3.7 [3.3–4.1]	0.130 ^d^

^a^ Mann–Whitney U. ^b^ Fisher’s Exact test. ^c^ Chi-Square. ^d^ Log-Rank test.

**Table 2 cancers-16-00356-t002:** Comparison between groups of high vs. low zonulin expression in tumoral specimens.

Variable		Low Zonulin (*n* = 11)	High Zonulin (*n* = 10)	*p*-Value
Age (years)		61.55 (SD = 14.00)	69.5 (SD = 7.66)	0.341 ^a^
Gender (male/female)		4:7	6:4	0.395 ^b^
Karnofsky < 70		2 (18.2%)	-	0.476 ^b^
Contrast enhancement	Periferic	6 (54.5%)	2 (20.0%)	0.183 ^b^
	Heterogeneous	5 (45.5%)	8 (80.0%)	
Contrast enhancement volume (cc)		12.91 (SD = 10.31)	28.13 (SD = 14.20)	0.020 ^a^
Edema volume (cc)		42.44 (SD = 19.14)	79.78 (SD = 43.13)	0.037 ^a^
Necrosis volume (cc)		6.50 (SD = 9.81)	10.85 (SD = 9.01)	0.152 ^a^
Resection	Partial	1 (9.1%)	1 (10.0%)	0.990 ^c^
	Subtotal	2 (18.2%)	2 (20.0%)	
	Total	8 (72.7%)	7 (70.0%)	
Ki67 (%)		26.2 (SD = 17.62)	28.6 (SD = 22.17)	0.880 ^a^
MGMT methylation		6 (54.5%)	4 (40.0%)	0.670 ^b^
Progression-free survival (months)		6.6 [2.4–10.9]	3.3 [0.0–6.5]	0.024 ^d^

^a^ Mann–Whitney U. ^b^ Fisher’s Exact test. ^c^ Chi-Square. ^d^ Log-Rank test.

## Data Availability

The datasets generated and analyzed during the current study are available upon reasonable request.

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
