# Peer review of "Evaluation of Zonulin Expression and Its Potential Clinical Significance in Glioblastoma"

_cancers, 2024, doi:10.3390/cancers16020356_

Round 1
Reviewer 1 Report
Comments and Suggestions for Authors
This manuscript analyzed the expression of Zonulin in GBM clinical specimens as well as GBM cell cultures. This study is significant with clear presentation of clinical data and Zonulin expression levels. I have three concerns as below.
1. The Zonulin western blotting results should be shown in the manuscript, including patient-derived tumor samples and GBM cell cultures. Now, the Zonulin western blotting of GBM cells was only shown in the supplemental results. Western blotting results of patient samples are missing.
2. The authors used three GBM cell lines that have been established and maintained in serum-containing medium. It is unclear if culturing these cell lines can convert them back to GBM stem cell (GSC) state. If so, GSC markers may need to be shown. Otherwise, the conclusion may need to be modified to show that GBM cell lines in serum-free GSC culture condition show enhanced Zonulin expression.
3. It is unclear if GBM cells in standard culture condition express Zonulin, based on the western blotting result shown in Supplementary Fig. 2.
Author Response
- The Zonulin western blotting results should be shown in the manuscript, including patient-derived tumor samples and GBM cell cultures. Now, the Zonulin western blotting of GBM cells was only shown in the supplemental results. Western blotting results of patient samples are missing.
Thank you very much for this observation. We have included the western blot of the cell lines in Figure 3. The western blot figure of the patients is of lower quality; hence, we have included it in the supplementary material (Supplementary Figure 2).
- The authors used three GBM cell lines that have been established and maintained in serum-containing medium. It is unclear if culturing these cell lines can convert them back to GBM stem cell (GSC) state. If so, GSC markers may need to be shown. Otherwise, the conclusion may need to be modified to show that GBM cell lines in serum-free GSC culture condition show enhanced Zonulin expression.
Thank you for this consideration. The intrinsic ability of glioma stem cells (GSCs) to form spheroid (tumorspheres) cultures emerges as a distinctive feature. The spontaneous and sustained formation of tumorspheres represents a key phenotype associated with GSCs, reflecting their unique ability to maintain self-renewal and tumorigenic capacity. This behavior in tumorspheres not only underscores the plasticity and self-renewing capability of GSCs but also serves as a relevant indicator in characterizing these glioma stem cells in experimental studies. We agree with you that the use of specific surface markers could contribute to a better characterization of these cells. In our case, we have determined the expression of CD133 for confirmation. We have not incorporated this information into the manuscript to avoid overloading it with details or confusing the reader with additional determinations that deviate from our main objective.
- It is unclear if GBM cells in standard culture condition express Zonulin, based on the western blotting result shown in Supplementary Fig. 2.
Thank you for this note. Zonulin was also expressed in glioblastoma cell lines cultured under standard conditions. Although a distinct band may not be visible, the quantification software exhibits high sensitivity, enabling proper quantification of pixel intensity.
Reviewer 2 Report
Comments and Suggestions for Authors
Repossi et al presented the clinical potential of the expression of the protein named Zonulin in glioblastoma, a brain tumour. Of the 34 glioblastoma patients, samples from 21 patients were obtained for zonulin expression and relevant data were collected using suitable methods. Further, zonulin expression was analyzed in vitro against three glioblastoma cell lines. Authors identified that the elevated zonulin levels in both tumor and serum of glioblastoma patients were linked to a poorer prognosis and showed an increase in glioma-stem cell-specific conditions. Thus, the authors explored the clinical significance of this protein through their research findings. The study is well-designed, and the results are presented in the manuscript, lucidly. based on the merit, the manuscript can be accepted in its present form.
Author Response
Thank you very much for the effort put into the review and thank you very much for your kind comments.
Reviewer 3 Report
Comments and Suggestions for Authors
The study "Zonulin Expression in Glioblastoma and its Potential Clinical Significance" authored by Repossi et al., is timely and is in the interest of a broad readership.
1. However, I would like to suggest that the author should establish the Zonulin expression level in the spheroid culture of the Primary GSCs line.
Comments on the Quality of English Language
NA
Author Response
- However, I would like to suggest that the author should establish the Zonulin expression level in the spheroid culture of the Primary GSCs line.
Thank you for this recommendation. This work is an initial approach to the potential role of zonulin in glioblastoma biology, and in this regard, we have observed higher levels of zonulin expression in 'induced' or 'non-primary' GSCs forming tumorspheres. It would be interesting to use the cell type you suggest in future studies, as well as GSCs directly derived from patient tumor samples. We have incorporated these suggestions into the manuscript.
Reviewer 4 Report
Comments and Suggestions for Authors
1) The title sounds like it is created for a review. Possible variant for the title of experimental article is “Evaluation of zonulin expression and its potential clinical significance in glioblastoma”. But it is up to you - you can leave original title or create another title.
2) English should be polished. “In-vitro” should be written without a hyphen (row 73, 85 etc.); “Glioma-stem cell” should be written without a hyphen (row 27, 37 etc.); misprint “immunoturbidimery” (row 96). I`m not sure that “tumoral probes” is correct term; alternative is “tumor specimens”. Also, the term “circulating levels” can be changed on “serum levels” or “in serum” etc.; it isn`t clear why you evade the word “serum”. Also, what means “The disruption of zonulin” (row 56)?
3) You can use the term “glioma stem cells” only if you made in vivo xenotransplantation studies in order to prove that your method of cancer cell treatment results in obtaining tumorigenic properties by glioma cells. Otherwise, you can use the term “tumorsphere” or “tumorsphere associated with glioma stem cells” with appropriate reference on the study in which it was shown that tumorspheres always possess exclusive tumorigenic properties. Also, instead of the term “glioma-stem cell (GSC) specific conditions”? it will be more correct to use the term “serum-free tumorsphere media”. If it is very important for you to use the terms “glioma stem cells” or “glioma-stem cell (GSC) specific conditions” you should explain, why do you think that these terms are appropriate in your situation, with references on other studies or another relevant information (several sentences with references).
4) It is not indicated in the Supplementary figure 2 what protein was detected. You mention Supplementary figure 3 (row 195) but there is no such figure in Supplementary file. There is no information in the legend of the Figure 2 what indicates red font. Please, check all Tables and Figures regarding their numeration, mentioning in text, and description in legend.
Comments on the Quality of English Language
English is easy to understand but it seems that minor editing of English language is required.
Author Response
- The title sounds like it is created for a review. Possible variant for the title of experimental article is “Evaluation of zonulin expression and its potential clinical significance in glioblastoma”. But it is up to you - you can leave original title or create another title.
Thank you very much for your suggestion. We have modified the title to clearly reflect the type of study we have conducted.
- English should be polished. “In-vitro” should be written without a hyphen (row 73, 85 etc.); “Glioma-stem cell” should be written without a hyphen (row 27, 37 etc.); misprint “immunoturbidimery” (row 96). I`m not sure that “tumoral probes” is correct term; alternative is “tumor specimens”. Also, the term “circulating levels” can be changed on “serum levels” or “in serum” etc.; it isn`t clear why you evade the word “serum”. Also, what means “The disruption of zonulin” (row 56)?
Thank you very much for your suggestions. We have taken them all into account and have modified the text accordingly.
- You can use the term “glioma stem cells” only if you made in vivo xenotransplantation studies in order to prove that your method of cancer cell treatment results in obtaining tumorigenic properties by glioma cells. Otherwise, you can use the term “tumorsphere” or “tumorsphere associated with glioma stem cells” with appropriate reference on the study in which it was shown that tumorspheres always possess exclusive tumorigenic properties. Also, instead of the term “glioma-stem cell (GSC) specific conditions”? it will be more correct to use the term “serum-free tumorsphere media”. If it is very important for you to use the terms “glioma stem cells” or “glioma-stem cell (GSC) specific conditions” you should explain, why do you think that these terms are appropriate in your situation, with references on other studies or another relevant information (several sentences with references).
Thank you for this suggestion. We have chosen to replace the term 'glioma stem cell' with 'tumorsphere'.
- It is not indicated in the Supplementary figure 2 what protein was detected. You mention Supplementary figure 3 (row 195) but there is no such figure in Supplementary file. There is no information in the legend of the Figure 2 what indicates red font. Please, check all Tables and Figures regarding their numeration, mentioning in text, and description in legend.
Thank you for your comments. We have reviewed the captions of all tables and figures in the manuscript. Regarding Supplementary Figure 2, we have ultimately included it in the main text (Figure 3). Concerning Supplementary Figure 3, there was a typographical error, as it corresponds to Supplementary Figure 1; we have already corrected it in the text. Finally, regarding the information in Figure 2, we have specified the meaning of the red color appearing in part of the figure text.
Round 2
Reviewer 3 Report
Comments and Suggestions for Authors
The article is very comprehensive and of the interest of the broader readership. The Authors have addressed the previous comments in the current manuscript.
Comments on the Quality of English LanguageNA